# Scar-Free Laparoscopy in BRCA-Mutated Women

**DOI:** 10.3390/medicina58070943

**Published:** 2022-07-17

**Authors:** Stefano Restaino, Angelo Finelli, Giulia Pellecchia, Anna Biasioli, Jessica Mauro, Carlo Ronsini, Monica della Martina, Martina Arcieri, Luigi Della Corte, Felice Sorrentino, Lorenza Driul, Giuseppe Vizzielli

**Affiliations:** 1Department of Maternal and Child Health, Obstetrics and Gynecology Clinic, University-Hospital of Udine, 33100 Udine, Italy; biasioli.anna@gmail.com (A.B.); monica.dellamartina@asufc.sanita.fvg.it (M.d.M.); lorenza.driul@uniud.it (L.D.); giuseppevizzielli@yahoo.it (G.V.); 2Department of Gyneacology and Obstetric Ulss2 Marca-Trevigiana Oderzo, Via Luzzatti Luigi 45, 31046 Oderzo, Italy; 3Department of Biomedical, Dental, Morphological and Functional Imaging Science, University of Messina, 98122 Messina, Italy; giulia.pellecchia95@gmail.com (G.P.); martina.arcieri89@gmail.com (M.A.); 4Department of Medicine, Obstetrics and Gynecology Clinic, University of Udine, 33100 Udine, Italy; jessica.mauro@me.com; 5Department of Woman, Child and General and Specialized Surgery, University of Campania “Luigi Vanvitelli”, 81100 Naples, Italy; carlo.ronsini@unicampania.it; 6Department of Neuroscience, Reproductive Sciences and Dentistry, School of Medicine, University of Naples, 80138 Naples, Italy; dellacorte.luigi25@gmail.com; 7Department of Medical and Surgical Sciences, Institute of Obstetrics and Gynecology, University of Foggia, 71122 Foggia, Italy; felice.sorrentino.1983@gmail.com

**Keywords:** BRCA, oncology, laparoscopy, scar free, percutaneous

## Abstract

*Background and Objectives*: BRCA 1 and 2 mutations have a cumulative risk of developing ovarian cancer at 70 years of 41% and 15%, respectively, while a cumulative risk of breast cancer by 80 years of age was 72% for BRCA1 mutation carriers and 69% for BRCA2 mutation carriers. The NCCN recommends risk-reducing salpingo-oophorectomy (RRSO), typically between 35 and 40 years, and upon completion of childbearing in BRCA1 mutation, while it is reasonable to delay RRSO for management of ovarian cancer risk until age 40–45 years in patients with BRCA2. In recent years there have been two main lines of evolution in laparoscopy. The former concerning the development of a single-site laparoscopic and the latter concerning the miniaturisation of laparoscopic instruments (mini/micro-laparoscopy). *Materials and Methods*: In this case report, we show our experience in prophylactic adnexectomy, on a mutated-BRCA patient, using the MiniLap^®^ percutaneous surgical system. *Results*: This type of technique is safe and effective and does not require a particular learning curve compared to single-port laparoscopy. *Conclusions***:** The considerable aesthetic advantage of the scars, we believe, albeit to a lesser extent, is useful to find in these patients burdened by an important stress load.

## 1. Introduction

BRCA 1 and 2 mutations have a cumulative risk of developing ovarian cancer at 70 years of 41% and 15%, respectively, while a cumulative risk of breast cancer by 80 years of age was 72% for BRCA1 mutation carriers and 69% for BRCA2 mutation carriers [1]. The NCCN recommends risk-reducing salpingo-oophorectomy (RRSO), typically between 35 and 40 years [2]. Approximately 16% of women with a BRCA1 or BRCA2 mutation experience distress levels comparable to those of women after a cancer diagnosis, and the lowest level of distress was for women who had risk-reducing surgery [3]. This high level of distress is linked to that for breast cancer screening; there are no screening tests that have been deemed effective in improving the detection of ovarian or fallopian tube cancer [4] and the currently available screening modalities, including transvaginal ultrasound and CA-125, have not been shown to reduce mortality related to ovarian cancer [5]. The breast cancer surveillance program includes: annual mammography, magnetic resonance imaging (MRI) and ultrasound; clinical examination every 6 months from 25 years of age, while in ovarian cancer, surveillance is conducted by integrating pelvic examination, transvaginal ultrasound and CA-125 assessment every 6 months, and these recommendations can have a very deleterious impact on the psychology of affected patients and affect many aspects of life due to the need for many outpatient hospitalizations. Risk-reducing surgery decreases cancer risk and worries in women with BRCA mutations [6]. The last decade has been characterized by an enormous evolution in the field of minimally invasive surgery, focusing its development mainly on two strategies: reducing the number of instruments or their size [7]. The first surgery reported with 3 mm instruments was a laparoscopic cholecystectomy in 1996, which has since spread to all surgical fields [8]. When women are offered the possibility of preventive interventions, general satisfaction with the decision is generally reported [9].

In this case report, we show our experience regarding prophylactic annessiectomy using the MiniLap^®^ percutaneous surgical system.

## 2. Materials and Methods

A 70-year-old woman PARA 102, body-mass index 28 (BMI, kg/m^2^) came to our centre after a genetic examination of susceptibility to the BRCA1 gene. Patient in oncological follow-up for breast cancer treated in 2017 with QUART (quadrantectomy with axillary dissection followed by electron beam radiation therapy) and sentinel lymph node, and subsequently with radiotherapy and letrozole therapy from 2018. The woman was simultaneously affected by essential hypertension and hypercholesterolemia. Menopausal period at 47 years with only 3 months of HRT (hormone replacement therapy), the patient denies AUB (Abnormal Uterine Bleeding). Previous major abdominal surgery only a laparoscopic cholecystectomy. The patient underwent routine blood tests, tumour markers and level II gynaecological ultrasound as a pre-operative evaluation. Oncological markers were CEA 0.4 ng/mL; CA 19.9 4.4 UI/mL, CA 15.3 13.4 UL/mL, CA 125 6.8 UL/mL. All pre-operative tests were negative for suspicion of an ovarian cancer. The planned and performed surgery was a laparoscopic adnexectomy using the MiniLap^®^ percutaneous surgical system.

The patient, under general anaesthesia, was positioned in the dorsal lithotomy position with both legs supported in stirrups with a Trendelenburg tilt and arms alongside the body. Four sterile trocars were used. A 10 mm port was inserted at the umbilicus for the telescope (0° high-definition telescope) using Hasson technique. In this case, the open technique was chosen for laparoscopic access as it is more familiar to the first surgeon. One additional 5 mm port was placed under direct vision after the pneumoperitoneum was achieved along the midline at least 8 cm from the umbilicus. Two MiniLap^®^ percutaneous alligator grasper (2.8 mm) were inserted in the right and left lower abdomen medial to the right obliterated umbilical artery and lateral to the inferior epigastric vessels at 1.5 cm above the anterior superior iliac spine (Figure 1 and Figure 2).

## 3. Results

The total operating time was 30 min and blood loss volume was minor (less than 50 cc). The hospitalization stay was 24 h. There were **no intra and** postoperative complications within 30 days. **The patient was discharged on the first day.** After 6 months, the patient was considered more than satisfied with the aesthetic outcome of the scars.

## 4. Discussion

For women with hereditary breast and ovarian cancer, the choice to undergo preventive surgery (mastectomy or bilateral salpingo-oophorectomy) is particularly harmful, both from a mental and physical point of view, as these, more or less, modify those anatomical parts that in society are characteristic of the feminine.

Domchek et al. [10] demonstrated that risk-reducing salpingo-oophorectomy could reduce the risk of ovarian cancer by 96% and breast cancer by 50–75% in unaffected women. Further, Powell et al. [6] described how risk-reducing surgery decreases worry in women with BRCA mutations, linked to the fact that the currently available screening modalities, including transvaginal ultrasound and CA-125, have not been shown to reduce mortality related to ovarian cancer [5] and to be effective in improving detection of ovarian or fallopian tube cancer [4]. D’Alonso et al. [11] reported 89% of patients’ satisfaction with risk-reducing salpingo-oophorectomy and of these, 63% would recommend it to a friend with a family risk of ovarian cancer.

The lower kinetic force required for the introduction of <5 mm caliber trocars ensures a safer and more controlled entry than conventional laparoscopic trocars. Recently, a ultra-minimally invasive concept has been reported using laparoscopic instruments that are inserted percutaneously without trocars [12], which are aimed to reduce surgical trauma, stress response, incision-related complications and hospital stay, and to provide a better cosmetic result [13]. Theoretically, mini instruments reduce the rate of incisional hernias and the extent of incisional trauma at the umbilical site and the abdominal wall vessels. One of the first bilateral adnexectomies via 2 mm ancillary instruments was reported by Ghezzi et al. [14]. Fanfani et al. reported that mini laparoscopy had a reduced postoperative pain with less requirement for analgesia when compared to conventional laparoscopy [15]. Further, Fanfani et al. [16] reported that the enhanced cosmetic results obtained through a minimally invasive approach also improve physical and psychological well-being, and younger women opted for this approach because of the possible reduced postoperative scarring. Gueli Alletti et al. [17], more recently, reported, as their postoperative pain evaluation and cosmetic outcomes, further support for the excellent scarless and painless effect of a percutaneously inserted instrument.

Ghezzi et al. [18] reported no significant differences between the mini-laparoscopic radical hysterectomy group and the conventional laparoscopic radical hysterectomy group in terms of operative time, blood loss, lymph node yield and complication rate, but with a shorter hospital stay. This can lead to an important cost-effectiveness ratio in the choice of instruments < 5 mm. One of the limitations of this technology, as Fanfani et al. [16] reported, is that the 3 mm bipolar was inadequate to control the bleeding of ruptured uterovaginal vessels and the absence of powerful bipolar energy still represents a significant limitation in the mini-laparoscopic approach, leading, in many cases, to the use of a 5 mm access for the introduction of a suitable coagulation instrument.

## 5. Conclusions

We support the use of this type of technique, as it does not require a particular learning curve compared to single-port laparoscopy, as reported by Misirlioglu et al. [19]. The considerable aesthetic advantage of the scars, we believe, albeit to a lesser extent, is useful to find in these patients burdened by an important stress load.

## Figures and Tables

**Figure 1 medicina-58-00943-f001:**
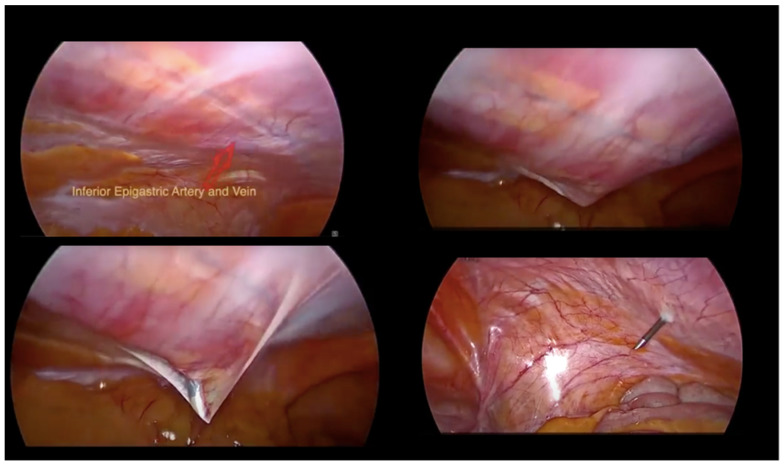
MiniLap^®^ skin accesses.

**Figure 2 medicina-58-00943-f002:**
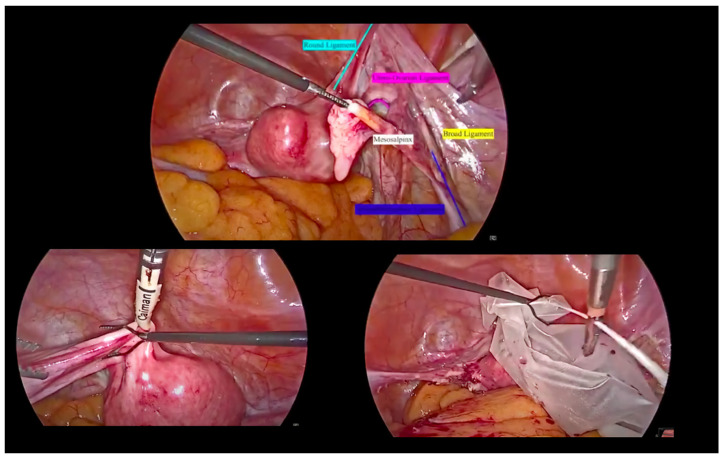
Step-by-step procedure.

## Data Availability

Not applicable.

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
