# Peer review of "Scar-Free Laparoscopy in BRCA-Mutated Women"

_medicina, 2022, doi:10.3390/medicina58070943_

Round 1

Reviewer 1 Report

1.      MiniLap Percutaneous Surgical System was inserted without trocar lead to fewer trauma, risk trocar side bleeding and smaller scar. But I am concerning about that this case use additional 5 mm port. Now that there are minipolar percutaneous electrosurgical probes instrument, why didn’t you use two MiniLap® instrument?

2.      This paper didn’t mention about cost effect of these minimal invasive laparoscopic procedure. If this paper can provide about more cost effectiveness of miniLap® compared than with traditional bipolar instrument and other minimal invasive laparoscopic instrument, it will be better.

3.      Some comparison between scar-free laparoscopy and traditional laparoscopy may need to be mentioned inside the discussion.

4.      If the author can provide the post- miniLap® procedure operative scar in the figure, that will be more sound and clear for the reader. 

Author Response

  1. MiniLap Percutaneous Surgical System was inserted without trocar lead to fewer trauma, risk trocar side bleeding and smaller scar. But I am concerning about that this case use additional 5 mm port. Now that there are minipolar percutaneous electrosurgical probes instrument, why didn’t you use two MiniLap® instrument?

We thank the Reviewer for the comments. Currently, this type of instrument does not incorporate electro-surgeon. For the type of procedure, it is necessary to be able to coagulate and cut, so a 5 mm accessory port is necessary.

  1. This paper didn’t mention about cost effect of these minimal invasive laparoscopic procedure. If this paper can provide about more cost effectiveness of miniLap® compared than with traditional bipolar instrument and other minimal invasive laparoscopic instrument, it will be better.

we thank the reviewers for this comment. Unfortunately, as mentioned in the text, this is a case report to show our experience regarding prophylactic annexiectomy using the MiniLap ® percutaneous surgical system. We certainly take the suggestion to perform a case-control study and assess the impact on costs.

  1. Some comparison between scar-free laparoscopy and traditional laparoscopy may need to be mentioned inside the discussion.

We thank the Reviewer for the comments. They have been very important and enlightening. In the light of that we have modified the text.

  1. If the author can provide the post- miniLap® procedure operative scar in the figure, that will be more sound and clear for the reader. 

We have collected the reviewer's suggestion, and we have highlighted the scar more clearly.

Reviewer 2 Report

This case report on scar-free laparoscopy represents an interesting topic especially for risk-reducing surgeries.

However, there are several deficiencies:

- it is highly unusual, that a manuscript contains references in the abstract and statements with references and factual data should be moved to the manuscript main body

- the abstract itself is not informative. It is not understandable how many patients were evaluated (I understand from the main body that this was one) and the results are non-informative

- the introduction in this manuscript is very short and does not state the state of the art in our understanding of scar free laparoscopy

- the methods section is underdeveloped, using abbreviations which are not explained. Please explain them and explain the instruments used with photos in the form of a figure. patient characteristics should also be reported in results. Why was Hasson technique used for entry, as there have been meta-analyses published that there is no difference with closed entry? Expand.

- results: consider reporting the characteristics of the surgery and patient in a table

- discussion: please compare the different currently available approaches to laparoscopic surgery and compare this in light of your used technique. Currently I do not believe the report to be sufficiently connected to the discussion? What outcomes you measured are connected with previous reports? 

Author Response

We thank the Reviewer for the comments. They have been very important and enlightening. In the light of that we have modified the text.

Round 2

Reviewer 2 Report

The authors have revised the manuscript in some aspects. However, revision on giving more purpose to this manuscript has not been done, therefore it is still not clear what the overall contribution of this casereport is? Is it just describing the technique? Do you want to measure the patient outcomes?  There was no report on short-term or long term outcomes of the patient following this procedure. This would streghten the report.

Further more some aspects from the introduction and discussion are repetitive in content.

A minor point is also that the figures need to be understood as stand alone items - you should add sufficient descriptions to this.

Author Response

Dear Reviewer,

The authors have revised the manuscript in some aspects. However, revision on giving more purpose to this manuscript has not been done, therefore it is still not clear what the overall contribution of this casereport is? Is it just describing the technique? Do you want to measure the patient outcomes?  There was no report on short-term or long term outcomes of the patient following this procedure. This would streghten the report.

We thank the reviewers for this comment. Unfortunately, as mentioned in the text, this is a case report to show our experience regarding prophylactic annexiectomy using the MiniLap ® percutaneous surgical system.

There was no report on short-term or long term outcomes of the patient following this procedure. This would streghten the report.

We have collected the reviewer's suggestion, and we have highlighted that. We underline that some aspects were already emphasised in the text

Further more some aspects from the introduction and discussion are repetitive in content.

We thank the Reviewer for the comments. We modified it.

A minor point is also that the figures need to be understood as stand alone items - you should add sufficient descriptions to this.

In accordance with the reviewer’s observation, we try to modify it.